# Technical note: Posterior Uncertainty Estimation via a Monte Carlo Procedure Specialized for 4D-Var Data Assimilation

Michael Stanley[1], Mikael Kuusela[2], Brendan Byrne[3], and Junjie Liu[4]

[1, 2]Department of Statistics and Data Science, Carnegie Mellon University, Pittsburgh, PA, USA
[3, 4]Jet Propulsion Laboratory, California Institute of Technology, Pasadena, CA, USA

**Correspondence:** Michael Stanley (mcstanle@andrew.cmu.edu)

**Abstract.** Through the Bayesian lens of data assimilation, uncertainty on model parameters is traditionally quantified through the posterior covariance matrix. However, in modern settings involving high-dimensional and computationally expensive forward models, posterior covariance knowledge must be relaxed to deterministic or stochastic approximations. In the carbon flux inversion literature, Chevallier et al. (2007) proposed a stochastic method capable of approximating posterior variances of linear functionals of the model parameters that is particularly well-suited for large-scale Earth-system data assimilation tasks. This note formalizes this algorithm and clarifies its properties. We provide a formal statement of the algorithm, demonstrate why it converges to the desired posterior variance quantity of interest, and provide additional uncertainty quantification allowing incorporation of the Monte Carlo sampling uncertainty into the method's Bayesian credible intervals. The methodology is demonstrated using toy simulations and a realistic carbon flux inversion observing system simulation experiment.

**Keywords** — Bayesian inference; inverse problem; uncertainty quantification; carbon flux inversion; uncertainty on uncertainty; observing system simulation experiment

## 1 Introduction

Uncertainty quantification (UQ) for data assimilation (DA) tasks is often non-trivial, but scientifically paramount to their understanding and interpretation. Since DA broadly describes methods combining observations with a computational model of a physical system, a Bayesian framework is often sensible for inference on the model parameters, as the posterior distribution quantifies knowledge resulting from this combination. As such, Bayesian statistical models are regularly used as the UQ framework. For example, Bayesian procedures play a central role in the general idea of optimal estimation (Rodgers, 2000), the broad field of DA (Kalnay, 2003), and the more specific field of carbon flux estimation (Deng et al., 2014; Liu et al., 2016). Inference for DA tasks using this statistical framework is typically challenging due to high-dimensional settings (e.g., high-resolution spatiotemporal grids) and the computer model's implicit numerical definition of the physical system of interest, often requiring supercomputers and long compute times to run the relevant forward model. Prior and observation error distributions are often assumed to be Gaussian, yielding a Gaussian posterior distribution under a linear forward model. Although a Gaussian posterior can be exactly characterized by its mean vector and covariance matrix, the high-dimensionality makes dealing directly with the posterior covariance matrix intractable and the implicit computationally demanding forward model makes standard

traditional Bayesian computational techniques, such as Markov Chain Monte Carlo (MCMC) infeasible. The implicit posterior necessitates the development of computational methods that implicitly access it.

The challenge of high-dimensional DA can be confronted by using a variational approach such as four-dimensional variational data assimilation (4D-Var) (Kalnay, 2003) (mathematically equivalent to optimal estimation as detailed in Rodgers (2000)), which aims to find an optimal model parameter vector via numerical optimization rather than sampling from a Markov

chain. The 4D-Var approach is the DA focus of this technical note. The optimal model parameter vector minimizes a cost function balancing model error with observations and proximity to prior information about the unknown state. When the forward model can be run in adjoint mode to obtain the cost function gradient, the optimal state can be iteratively solved using a first-order optimization method. $CO_2$ flux inversion is a representative example of a high-dimensional DA task to which Bayesian modeling is applied and 4D-Var is used to compute estimated flux fields (Crowell et al., 2019; Enting et al., 1995; Gurney

et al., 2002). In this problem, estimates of net surface-atmosphere $CO_2$ fluxes are inferred from atmospheric $CO_2$ measurements, with fluxes and atmospheric measurements being related by a chemical transport model (the computational forward model). However, the relatively sparse atmospheric $CO_2$ observations underconstrain surface fluxes of $CO_2$, and regularization with prior information is the Bayesian approach to making the problem well-posed. These analyses have historically assimilated measurements of atmospheric $CO_2$ from a global network of flask and in situ measurements (Enting et al., 1995), but

more recent work (Byrne et al., 2023; Crowell et al., 2019; Deng et al., 2014; Houweling et al., 2015) has shifted to assimilating space-based column-averaged dry-air mole fractions, denoted $X_{CO_2}$, as observations availability has expanded since 2009. In these analyses, the prior and error distributions are typically assumed to be Gaussian and the forward model can be reasonably assumed linear in the net surface-atmosphere fluxes.

When the number of model parameters is low and a forward model run is inexpensive, it is possible to explicitly construct

the posterior covariance matrix. Successful examples of this approach date back at least to Vasco et al. (1993) in seismic tomography, where inversion is performed on 12,496 model parameters. However, more contemporary problems typically have orders of magnitude more parameters and substantially more expensive forward models, requiring other approaches to access posterior covariance matrix information. Once the discretization of the computational model is set, the dimensionality problem can be handled either by defining an approximate statistical model on a lower dimensional problem, or by working

in some subspace of the full-dimensional problem. A recent example of the first strategy is seen in Zammit-Mangion et al. (2022) in the WOMBAT inversion system which lowers the dimension of the statistical model via an intelligently chosen set of basis functions, facilitating MCMC. Alternatively, Petra et al. (2014) propose with Stochastic Newton MCMC (SN-MCMC) the possibility for MCMC in the full parameter space by using a low-rank approximation to the posterior covariance within the proposal distribution of a Metropolis–Hasting algorithm. Although WOMBAT and SN-MCMC are both MCMC-based,

WOMBAT assumes a linear forward model, while SN-MCMC does not, allowing it to characterize non-Gaussian posteriors. Staying with a linear forward model assumption, other approaches leverage low-rank posterior covariance approximations. Flath et al. (2011) develop a low-rank algorithm for approximating the posterior covariance by computing the leading eigenvalues and eigenvectors of a prior-conditioned Hessian matrix of the associated objective function (i.e., the log posterior). In a similar spirit, Kalmikov and Heimbach (2014) provide a derivative-based algorithm to compute leading Hessian eigenval-

ues and eigenvectors and extend the uncertainty quantification to quantities of interest in global ocean state estimation. The algorithms in both Flath et al. (2011) and Kalmikov and Heimbach (2014) rely upon the Lanczos method (Lanczos, 1950) for matrix-free computation of the low-rank approximation. Alternatively, Bousserez and Henze (2018) more recently proposed a low-rank approximation algorithm dependent upon the randomized SVD algorithm (Halko et al., 2011). All of the aforementioned methods can be grouped by their reliance upon some low-dimensional deterministic approximation.

In contrast, stochastic approximations of the posterior distribution rely neither upon pre-inversion dimension reductions nor low-rank matrix approximations, but rather generate ensembles of inversions using random generators. These approaches usually share fundamental model (e.g., linearity of the forward model in the model parameters) and observation error (e.g., Gaussian errors) assumptions. Although approaches like particle filtering allow for relaxation of these assumptions, they have mostly been successful for relatively low-dimensional problems and are in a nascent stage of applications to high-dimensional DA tasks (Doucet et al., 2000; Potthast et al., 2018; Maclean and Vleck, 2021). As such, there is still a great need for UQ methods for high-dimensional DA approaches relying upon those common assumptions. In carbon flux inversion, Chevallier et al. (2007) developed such a method to estimate the posterior variance of *functionals* of the flux field (i.e., maps from the flux field to the reals). The method uses the forward model, specified prior, and known observation error distributions in a particularly efficient manner. Broadly, the algorithm creates an ensemble of prior means and observation errors, sampling according to their respective distributions. For each ensemble member, it finds the maximum a posteriori (MAP) estimator, to which the functional is applied. Finally, it finds the empirical variance across the ensemble members to estimate the posterior variance of the functional. This method is well-suited for carbon flux estimation and 4D-Var-based DA UQ more generally for a few key reasons. First, each ensemble member is computationally independent, making the method parallelizable and hence offering a substantial computational benefit compared to sequential methods, such as MCMC. Second, although in general prior misspecification biases the posterior, the prior mean does not need to be correctly specified in order for the procedure to produce an unbiased estimator of the posterior variance. Third, the ensemble of inversions can flexibly produce UQ estimates for arbitrary functionals post hoc, as opposed to requiring the specification of a functional ahead of the analysis. Finally, since this method is more generally a Monte Carlo (MC) method for a Gaussian statistical model, the method's sampling uncertainty can be analytically characterized and accounted for in the final UQ estimate. The ability to easily characterize this uncertainty of the uncertainty stands in contrast to the difficulty in characterizing deterministic error of the aforementioned low-dimensional approaches.

Although Chevallier et al. (2007) appear to have been the first to develop this method, which was later applied in Liu et al. (2016), we are unaware of a formal statement or analysis of this algorithm. These previous works also did not quantify the algorithm's MC uncertainty. As such, the primary contributions of this paper are a rigorous formal statement of the algorithm, an analysis showing the convergence of its output to the true posterior quantity of interest, and uncertainty quantification of the algorithm itself so that the algorithm's sampling uncertainty can be accounted for in the final inference. Since this algorithm originated in the carbon flux inversion literature, we use carbon flux inversion as a running application example through this technical report. However, the approach is applicable to any 4D-Var DA implementation subject to the mathematical assumptions and considerations outlined in Section 2.

**Table 1.** Mathematical symbols and notation used herein (roughly in order of appearance).

| | | | |
|---|---|---|---|
| $\mathbf{c} \in \mathbb{R}^m$ | Parameter vector | $\mathbf{y} \in \mathbb{R}^n$ | Observation vector |
| $\boldsymbol{\mu} \in \mathbb{R}^m$ | Control vector | $f$ | Forward model |
| $\boldsymbol{\epsilon} \in \mathbb{R}^n$ | Observation Noise | $\mathbf{R}$ | Observation Noise Covariance |
| $\mathbf{A} \in \mathbb{R}^{n \times m}$ | Linear forward model | $\mathbf{c}^b \in \mathbb{R}^m$ | Prior expectation |
| $\mathbf{B} \in \mathbb{R}^{m \times m}$ | Prior covariance | $\pi(\mathbf{c} \mid \mathbf{y})$ | Posterior density |
| $\mathbf{I}_m \in \mathbb{R}^{m \times m}$ | $m \times m$ identity matrix | $\boldsymbol{\Sigma} \in \mathbb{R}^{m \times m}$ | Posterior covariance |
| $\boldsymbol{\alpha} \in \mathbb{R}^m$ | Posterior expectation | $\boldsymbol{\delta} \in \mathbb{R}^m$ | Posterior physical quantity expectation |
| $\boldsymbol{\Gamma} \in \mathbb{R}^{m \times m}$ | Posterior physical quantity covariance | $(\mathbf{c}_k, \mathbf{y}_k) \in \mathbb{R}^m \times \mathbb{R}^n$ | $k$th MC sample |
| $\mathbf{c}_e \in \mathbb{R}^m$ | MC Prior mean expectation | $\mathbf{y}_e \in \mathbb{R}^m$ | MC control observation |
| $\boldsymbol{\Sigma}_{\mathbf{c}_{MAP}^k} \in \mathbb{R}^{m \times m}$ | MC MAP estimator covariance | $\mathbf{h} \in \mathbb{R}^m$ | Functional of interest |
| $\overline{\varphi} \in \mathbb{R}$ | Mean MC functional value | $\hat{\sigma}_\varphi^2 \in \mathbb{R}_+$ | Empirical functional variance |
| $\chi_{M-1}^2$ | Chi-squared distribution with $M-1$ dof | $\chi_{M-1,\alpha/2}^2$ | Chi-squared $(\alpha/2)$-quantile |
| $\alpha \in (0,1)$ | Frequentist confidence level | $\gamma \in (0,1)$ | Bayesian credible interval level |
| $L$ | Deflation factor for MC variance | $R$ | Inflation factor for MC variance |
| $\mathbf{z} \in \mathbb{R}^n$ | Non-biospheric $X_{CO_2}$ component | $\tilde{\mathbf{y}} \in \mathbb{R}^n$ | $X_{CO_2}$ Observations |
| $b^2 \in \mathbb{R}_+$ | Prior variance parameter | $\boldsymbol{\theta} = \mathbf{c} \circ \boldsymbol{\mu}$ | Physical quantity of interest (flux vector) |
| $\mathbf{a} \circ \mathbf{b}$ | Element-wise multiplication of $\mathbf{a}$ and $\mathbf{b}$ | $\mathbf{A}_{\boldsymbol{\mu}} \in \mathbb{R}^{n \times m}$ | Forward model with control flux $\boldsymbol{\mu}$ |

The rest of this paper is structured as follows. In Section 2, we fully describe the algorithm, present mathematical results proving its correctness, and derive deflation and inflation factors to apply to the estimated posterior uncertainty to quantify the MC uncertainty. Proofs of the mathematical results can be found in Appendix A. In Section 3, we provide two experimental demonstrations: the first is a low-dimensional problem in which we explicitly know the linear forward model and the second is a carbon flux observing system simulation experiment (OSSE) to which we applied this method to compute global monthly flux credible intervals along their MC uncertainty. Finally, we provide some concluding remarks in Section 4. For reference, all mathematical notation in order of appearance is collected in Table 1.

## 2 Monte Carlo Method Exposition, Analysis, and Uncertainty Quantification

### 2.1 The Bayesian 4D-Var Setup

The following equation describes the relationship between a parameter vector, $\mathbf{c} \in \mathbb{R}^m$, that we wish to optimize and an observation vector $\mathbf{y} \in \mathbb{R}^n$:

$$\mathbf{y} = f(\mathbf{c}; \boldsymbol{\mu}) + \boldsymbol{\epsilon}, \quad \boldsymbol{\epsilon} \sim \mathcal{N}(\mathbf{0}, \mathbf{R}), \tag{1}$$

where $f(\cdot; \cdot)$ is the forward model mapping from parameter space to observation space, $\boldsymbol{\mu} \in \mathbb{R}^p$ is a control vector of parameters that remain fixed during the optimization and $\mathbf{R} \in \mathbb{R}^{n \times n}$ is the observation covariance matrix. To streamline notation, we

denote henceforth the forward model using only the parameter vector as input, i.e., $f(\mathbf{c})$. To regularize the problem and provide uncertainty quantification on the estimated parameter vector, $\mathbf{c}$ in Equation (1) is given a Gaussian prior distribution, yielding the following Bayesian generative model,

$$\mathbf{c} \sim \mathcal{N}(\mathbf{c}^b, \mathbf{B}), \tag{2}$$

$$\mathbf{y} \mid \mathbf{c} \sim \mathcal{N}(f(\mathbf{c}), \mathbf{R}), \tag{3}$$

where $\mathbf{c}^b \in \mathbb{R}^m$ is the prior mean and $\mathbf{B}$ is the prior covariance matrix. Finding the mode of the posterior $\mathbf{c} \mid \mathbf{y}$ defined by Equations (2) and (3) is the objective of 4D-Var data assimilation, a method simultaneously optimizing the parameter vector over all time steps, which has been applied to a wide range of applications in Earth sciences (Deng et al., 2014; Kalmikov and Heimbach, 2014; Liu et al., 2016). 4D-Var can be regarded as a least-squares optimization with an $\ell_2$ regularizer (ridge regression), or, equivalently, as maximum a posteriori (MAP) estimation in the Bayesian paradigm. This connection means that the 4D-Var optimization is connected to the posterior resulting from the prior and likelihood in Equations (2) and (3). From the Bayesian perspective, the 4D-Var cost function $F(\mathbf{c})$ is the negative log-posterior density of the scaling factors given the observations,

$$F(\mathbf{c}) = -\log(\pi(\mathbf{c} \mid \mathbf{y})) = \frac{1}{2} \left(\mathbf{c} - \mathbf{c}^b\right)^\top \mathbf{B}^{-1} \left(\mathbf{c} - \mathbf{c}^b\right) + \frac{1}{2} \left(\mathbf{y} - f(\mathbf{c})\right)^\top \mathbf{R}^{-1} \left(\mathbf{y} - f(\mathbf{c})\right) + C, \tag{4}$$

where $C \in \mathbb{R}$ is a normalizing constant for the posterior distribution and $\pi(\mathbf{c} \mid \mathbf{y})$ denotes the posterior density. Thus, finding the MAP estimator, i.e., the $\mathbf{c}$ that maximizes the posterior density, is equivalent to finding the vector $\mathbf{c}$ that minimizes the 4D-Var cost function. As such, uncertainty quantification can be handled through the covariance matrix of the posterior, which we denote as $\mathbf{\Sigma} = \mathrm{Cov}\left(\mathbf{c} \mid \mathbf{y}\right)$.

To facilitate exposition, we assume the forward model is linear, i.e., $f(\mathbf{c}) = \mathbf{A}\mathbf{c}$, though we address strategies to apply the Monte Carlo procedure described in Section 2.2 to nonlinear forward models in Section 2.2.1. As such, we assume the following linear Bayesian model,

$$\mathbf{c} \sim \mathcal{N}(\mathbf{c}^b, \mathbf{B}), \tag{5}$$

$$\mathbf{y} \mid \mathbf{c} \sim \mathcal{N}(\mathbf{A}\mathbf{c}, \mathbf{R}), \tag{6}$$

on which the primary analysis of this technical note is performed. The linearity assumption is not only helpful to expose the validity of the Monte Carlo algorithm under consideration, but it is also valid in carbon flux inversion, our primary application of interest. We discuss the necessary adjustments to apply this approach to carbon flux inversion in Section 3.2. Assuming linearity, Gaussian prior, and Gaussian likelihood ensures the equivalence between the posterior mode and expectation, and hence optimizing (4) is equivalent to obtaining the posterior expectation. We emphasize that the matrix $\mathbf{A}$ is not explicitly available to us, but is instead implicitly defined by a computational model. Thus, even though the linear assumption allows explicit analytical interpretation, we are not able to interact with the matrix in arbitrary ways in practice.

Assuming linearity of the forward model, the posterior mean (mode) and covariance of $\mathbf{c}$ are analytically tractable. Hence, $\mathbf{c} \mid \mathbf{y} \sim \mathcal{N}(\boldsymbol{\alpha}, \boldsymbol{\Sigma})$, where by the posterior mean and covariance Equations 4.3 and 4.7 in Rodgers (2000) we have:

$$\boldsymbol{\Sigma} = \left(\mathbf{A}^\top \mathbf{R}^{-1} \mathbf{A} + \mathbf{B}^{-1}\right)^{-1}, \tag{7}$$

$$\boldsymbol{\alpha} = \boldsymbol{\Sigma} \left(\mathbf{A}^\top \mathbf{R}^{-1} \mathbf{y} + \mathbf{B}^{-1} \mathbf{c}^b\right). \tag{8}$$

Note, $\boldsymbol{\alpha}$ is also the MAP estimator of $\mathbf{c}$. Since the forward model $\mathbf{A}$ is only known implicitly via a computer simulator, making direct use of Equations (7) and (8) is intractable. Instead, the 4D-Var cost function in Equation (4) is typically minimized using the L-BFGS-B algorithm (Byrd et al., 1995) with the cost function gradient computed numerically using the adjoint of $f$ (Henze et al., 2007). L-BFGS-B can often find a reasonable approximation of the posterior mean / mode $\boldsymbol{\alpha}$ in a handful of iterations. We now describe the Monte Carlo method that provides an approach for uncertainty quantification of $\boldsymbol{\alpha}$ despite the analytical intractability of the posterior covariance $\boldsymbol{\Sigma}$.

## 2.2 The Monte Carlo Procedure

To execute the Monte Carlo procedure introduced in Chevallier et al. (2007), we generate $M$ ensemble members. For each $k = 1, 2, \ldots, M$, we sample a new prior mean $\mathbf{c}_k$ and new observation $\mathbf{y}_k$ as follows:

$$\mathbf{c}_k \stackrel{\text{i.i.d.}}{\sim} \mathcal{N}(\mathbf{c}_e, \mathbf{B}), \tag{9}$$

$$\mathbf{y}_k \stackrel{\text{i.i.d.}}{\sim} \mathcal{N}(\mathbf{y}_e, \mathbf{R}), \tag{10}$$

where $\mathbf{c}_e$ is a chosen prior expectation to perturb and $\mathbf{y}_e \in \mathbb{R}^n$ is a chosen control observation to perturb. Chevallier et al. (2007) operated in a simulation setting where they knew the true parameter, which we call $\mathbf{c}_{clim}$ following their notation. As such, they set $\mathbf{c}_e = \mathbf{c}_{clim}$ and $\mathbf{y}_e = \mathbf{A}\mathbf{c}_{clim}$. When using real data, $\mathbf{c}_{clim}$ is not known and so it is sensible to set this expectation using a physically reasonable value (for example, see Section 3.2 for how to set these expectations in more recent carbon flux inversion settings). However, as we show below, the choice of the expectations in (9) and (10) does not particularly matter under the linear-Gaussian assumption (see Equation (12)), as long as $\text{Cov}(\mathbf{B}^{-1}\mathbf{c}_k, \mathbf{A}^\top \mathbf{R}^{-1}\mathbf{y}_k) = \mathbf{0}$. So, each Monte Carlo iteration involves sampling a pair $(\mathbf{c}_k, \mathbf{y}_k) \in \mathbb{R}^m \times \mathbb{R}^n$ according to Equations (9) and (10).

The MAP estimator from Equation (8) corresponding to prior mean $\mathbf{c}_k$ and observation $\mathbf{y}_k$ is analytically tractable for each ensemble member when $\mathbf{A}$ is explicitly known:

$$\mathbf{c}_{MAP}^k = \boldsymbol{\Sigma} \left(\mathbf{A}^\top \mathbf{R}^{-1} \mathbf{y}_k + \mathbf{B}^{-1} \mathbf{c}_k\right). \tag{11}$$

Since $\mathbf{c}_{MAP}^k$ is a linear function of two Gaussian random variables, it is also Gaussian. Therefore, linear combinations of it are also Gaussian, which will be a useful fact in the analysis performed in Section 2.3. The covariance matrix of this MAP

estimator, henceforth denoted as $\boldsymbol{\Sigma}_{\mathbf{c}_{MAP}^k}$, is

$$
\begin{aligned}
\boldsymbol{\Sigma}_{\mathbf{c}_{MAP}^k} &= \boldsymbol{\Sigma} \operatorname{Cov} \left[ \mathbf{A}^\top \mathbf{R}^{-1} \mathbf{y}_k + \mathbf{B}^{-1} \mathbf{c}_k \right] \boldsymbol{\Sigma}^\top \\
&= \boldsymbol{\Sigma} \left( \operatorname{Cov} \left[ \mathbf{A}^\top \mathbf{R}^{-1} \mathbf{y}_k \right] + \operatorname{Cov} \left[ \mathbf{B}^{-1} \mathbf{c}_k \right] \right) \boldsymbol{\Sigma} \\
&= \boldsymbol{\Sigma} \left( \operatorname{Cov} \left[ \mathbf{A}^\top \mathbf{R}^{-1} \mathbf{y}_k \right] + \mathbf{B}^{-1} \right) \boldsymbol{\Sigma} \\
&= \boldsymbol{\Sigma} \left( \mathbf{A}^\top \mathbf{R}^{-1} \operatorname{Cov} \left[ \mathbf{y}_k \right] \mathbf{R}^{-1} \mathbf{A} + \mathbf{B}^{-1} \right) \boldsymbol{\Sigma} \\
&= \boldsymbol{\Sigma} \left( \mathbf{A}^\top \mathbf{R}^{-1} \mathbf{A} + \mathbf{B}^{-1} \right) \boldsymbol{\Sigma} = \boldsymbol{\Sigma},
\end{aligned}
\tag{12}
$$

where the covariance decomposition in line two follows from $\operatorname{Cov}(\mathbf{B}^{-1}\mathbf{c}_k, \mathbf{A}^\top \mathbf{R}^{-1}\mathbf{y}_k) = \mathbf{0}$ and the final equality follows from $\boldsymbol{\Sigma}^{-1} = \left( \mathbf{A}^\top \mathbf{R}^{-1} \mathbf{A} + \mathbf{B}^{-1} \right)$. Thus, the covariance matrix of the Monte Carlo ensemble MAP estimators is equal to the desired posterior covariance $\boldsymbol{\Sigma}$. We also note that using the expectation choices from Chevallier et al. (2007) of $\mathbf{c}_e = \mathbf{c}_{clim}$ and $\mathbf{y}_e = \mathbf{A}\mathbf{c}_{clim}$ means that the $\mathbb{E}[\mathbf{c}_{MAP}^k] = \mathbf{c}_{clim}$.

The above covariance equality is the key fact allowing this method to work, as it allows us to compute an empirical estimate of the posterior covariance by sampling from two unconditional distributions and solving the 4D-Var objective. To the best of our knowledge, proof of this equality has not appeared in previous literature on this method. However, there is a similar method used in the spatial statistics literature to sample from conditional random fields as shown in Chapter 3, Section 3.6.2 of Cressie (1993) and Chapter 7, Section 7.3.1 of (Chiles and Delfiner, 2012). As described in Chiles and Delfiner (2012), this sampling method is able to sample from complex conditional random fields if one has access to a simulator with the same covariance as the distribution of interest. This matches the crucial requirement of the Monte Carlo procedure mentioned above and suggests a potential connection between these two sampling procedures.

Since the linear forward model is not explicitly available in most 4D-Var scenarios, each ensemble member MAP estimator $\mathbf{c}_{MAP}^k$ must be obtained with an iterative optimization algorithm minimizing (4) with $\mathbf{c}_k$ and $\mathbf{y}_k$ as the prior mean and observation vectors. Once these ensemble members are obtained, we could in principle estimate the posterior covariance matrix $\boldsymbol{\Sigma}$ with the empirical covariance estimator $\widehat{\boldsymbol{\Sigma}}$ based on the Monte Carlo ensemble as follows,

$$
\widehat{\boldsymbol{\Sigma}} = \frac{1}{M-1} \sum_{k=1}^{M} \left( \mathbf{c}_{MAP}^k - \overline{\mathbf{c}} \right) \left( \mathbf{c}_{MAP}^k - \overline{\mathbf{c}} \right)^\top,
\tag{13}
$$

where $\overline{\mathbf{c}} = \frac{1}{M} \sum_{k=1}^{M} \mathbf{c}_{MAP}^k$. However, in practice, most data assimilation applications, like carbon flux inversion, are high dimensional, making direct interaction with these covariance matrices difficult. Indeed, accurate estimation of $\boldsymbol{\Sigma}$ using Equation (13) would require an enormously large Monte Carlo ensemble and would require storing and working with an $m \times m$ matrix, where $m \sim 10^5$ or larger. Fortunately, we often care about the variance of one-dimensional summaries of $\mathbf{c}$, such as the posterior variance of some linear combination of $\mathbf{c}$, as opposed to the full posterior covariance matrix. For instance, we might wish to estimate North American carbon fluxes over a particular month.

Obtaining quantities of the above type is mathematically implemented using a linear functional of the underlying high-dimensional parameter. That is, we wish to characterize the posterior of $\varphi(\mathbf{c}) = \boldsymbol{h}^\top \mathbf{c}$, where $\boldsymbol{h} \in \mathbb{R}^m$ contains weights necessary to aggregate the desired vector components. Hence, building off Equations (7) and (8), we obtain the posterior distribution

for the functional of interest:

$$\varphi(\mathbf{c}) \mid \mathbf{y} \sim \mathcal{N}(\boldsymbol{h}^\top \boldsymbol{\alpha}, \boldsymbol{h}^\top \boldsymbol{\Sigma} \boldsymbol{h}). \tag{14}$$

We wish to obtain the posterior variance of this functional. Define $\sigma_\varphi^2 = \mathrm{Var}(\varphi(\mathbf{c}) \mid \mathbf{y}) = \boldsymbol{h}^\top \boldsymbol{\Sigma} \boldsymbol{h}$. We could inefficiently estimate this using $\hat{\sigma}_\varphi^2 = \boldsymbol{h}^\top \hat{\boldsymbol{\Sigma}} \boldsymbol{h}$, but we wish to avoid working directly with the full empirical covariance matrix. The following algebraic steps provide a better alternative:

$$\hat{\sigma}_\varphi^2 = \boldsymbol{h}^\top \left( \frac{1}{M-1} \sum_{k=1}^{M} \left( \mathbf{c}_{MAP}^k - \bar{\mathbf{c}} \right) \left( \mathbf{c}_{MAP}^k - \bar{\mathbf{c}} \right)^\top \right) \boldsymbol{h} \tag{15}$$

$$= \frac{1}{M-1} \sum_{k=1}^{M} \boldsymbol{h}^\top \left( \mathbf{c}_{MAP}^k - \bar{\mathbf{c}} \right) \left( \mathbf{c}_{MAP}^k - \bar{\mathbf{c}} \right)^\top \boldsymbol{h} \tag{16}$$

$$= \frac{1}{M-1} \sum_{k=1}^{M} \left[ \boldsymbol{h}^\top \left( \mathbf{c}_{MAP}^k - \bar{\mathbf{c}} \right) \right]^2 \tag{17}$$

$$= \frac{1}{M-1} \sum_{k=1}^{M} \left( \varphi_k - \overline{\varphi} \right)^2, \tag{18}$$

where $\varphi_k = \boldsymbol{h}^\top \mathbf{c}_{MAP}^k$ and $\overline{\varphi} = \boldsymbol{h}^\top \bar{\mathbf{c}} = \frac{1}{M} \sum_{k=1}^{M} \varphi_k$. The above algebra shows that the posterior variance of the functional can be computed using the functionals of the Monte Carlo samples without having to form the full empirical covariance matrix. See Algorithm 1 for a succinct exposition of the above procedure. Notice that the functional does not need to be specified when creating the Monte Carlo ensemble. As long as the ensemble $\{\mathbf{c}_{MAP}^k\}_{k=1}^{M}$ is stored and made available to the end users, they may evaluate post-hoc the uncertainty of any functional that is of interest in their specific use-case.

### 2.2.1   Considerations for when $f$ is nonlinear and other critical assumptions

Our demonstration of this procedure's validity relies on several assumptions which we restate here to clarify and comment on the procedure's resilience to their violation. For the primary applications of interest where the forward model is not analytically tractable, this approach's feasibility relies upon efficient computation of the posterior expectation. Furthermore, proving this algorithm's validity relies upon the equivalence between the posterior covariance and the covariance of the ensemble members. We showed this equivalence by appealing to equations following from the linear forward model and Gaussian errors assumptions. Since relaxing the Gaussian assumption would completely change the 4D-Var objective function and this technical note is primarily about standard 4D-Var, we do not consider this relaxation. However, linearity is not necessary to use 4D-Var, which is one of the benefits of using such a variational approach. Although it is possible that covariance equivalence holds for nonlinear forward models, the linearity assumption is necessary in our demonstration, since we require the equivalence between the posterior expectation and MAP to show the covariance of the ensemble element.

There are at least two options for posterior covariance-based uncertainty quantification under a nonlinear forward model based on linearizing the forward model around a particular point in the parameter space. Linearizing the forward model around

---

**Algorithm 1** Monte Carlo Algorithm to Estimate Posterior Uncertainty in 4D-Var Data Assimilation

---

**Inputs**:

- $M \in \mathbb{N}$: Number of Monte Carlo samples.

- $\mathbf{R} \in \mathbb{R}^{n \times n}$: Observation error covariance built from observation uncertainties.

- $\mathbf{B} \in \mathbb{R}^m$: Parameter prior variance.

- $\mathbf{A} \in \mathbb{R}^{n \times m}$: Forward model mapping from parameter to observations space (note, this can be known either explicitly via a matrix or implicitly via a computer simulator).

- $\mathbf{c}_e \in \mathbb{R}^m$: Prior expectation.

- $\mathbf{y}_e \in \mathbb{R}^n$: Control observation.

- $\boldsymbol{h} \in \mathbb{R}^m$: Vector defining the functional of interest.

**Steps**:

1. Let $\mathcal{S}$ denote an array of length $M$ that will store the MAP estimators for each Monte Carlo sample.

2. For $k = 1, \ldots, M$:

   (a) Simulate $\mathbf{c}_k \sim \mathcal{N}(\mathbf{c}_e, \mathbf{B})$.

   (b) Simulate $\boldsymbol{y}_k \sim \mathcal{N}(\mathbf{y}_e, \mathbf{R})$.

   (c) Find MAP estimator $\mathbf{c}_{MAP}^k$. If $\mathbf{A}$ is known explicitly, it can be found using Equation (11). If $\mathbf{A}$ is known implicitly through a computational model, use a numerical optimizer (e.g., L-BFGS-B) to optimize a the 4D-Var cost function as defined in Equation (4).

   (d) $\mathcal{S}[k] \leftarrow \mathbf{c}_{MAP}^k$.

3. Estimate Posterior Functional Variance:

   (a) Compute the mean Monte Carlo sample functional: $\overline{\varphi} = \frac{1}{M} \sum_{k=1}^{M} \varphi_k$, where $\varphi_k = \boldsymbol{h}^\top \mathbf{c}_{MAP}^k$.

   (b) Compute the empirical posterior functional variance:

$$\hat{\sigma}_\varphi^2 = \frac{1}{M-1} \sum_{k=1}^{M} (\varphi_k - \overline{\varphi})^2.$$

---

a point $\mathbf{x_0}$ takes the form,

$$f(\mathbf{c}) = f(\mathbf{c_0}) + \nabla_\mathbf{c} f(\mathbf{c_0})(\mathbf{c} - \mathbf{c_0}), \tag{19}$$

where $\nabla_\mathbf{c} f(\mathbf{c_0}) \in \mathbb{R}^{n \times m}$ is the Jacobian of the forward model about $\mathbf{c_0}$, and effectively becomes the linear forward model under linearization. Following the terminology of Rodgers (2000), linearization is applicable for "nearly linear" and "moderately nonlinear" problems. Nearly linear problems are those for which "linearization about some prior state is adequate to

230 find a solution." (Rodgers, 2000) In this circumstance, one can linearize the forward model around a prior guess, $\mathbf{c}_0$, and proceed with 4D-Var and the Monte Carlo uncertainty quantification procedure as if the forward model is linear as described above. Moderately nonlinear problems are those "where linearization is adequate for the error analysis, but not for finding a solution." (Rodgers, 2000) As such, one can numerically obtain a solution, $\hat{\mathbf{c}}$, to the 4D-Var objective including the nonlinear forward model followed by a linearization around $\hat{\mathbf{c}}$. One can then perform the Monte Carlo procedure with the forward model
linearized about the solution to the 4D-Var objective.

## 2.3 Quantifying the Monte Carlo Uncertainty

Although Section 2.2 establishes the equality of the Monte Carlo MAP estimator ensemble member covariance to the posterior covariance (and therefore the equality of the Monte Carlo ensemble member functional variance to the posterior functional variance), we have not yet established that the empirical covariance matrix (and functional variance) converges in probability
to the true posterior covariance matrix (and functional variance). There are consistency results showing the empirical covariance matrix converging in probability to the true covariance matrix (see, for instance, Chapter 6 in Wainwright (2019)). However, since this application is primarily concerned with linear functionals of the form $\varphi(\mathbf{c}) = \boldsymbol{h}^\top \mathbf{c}$ as described in Section 2.2, we can appeal directly to the consistency of the sample variance as shown, for example, in Chapter 5 in Casella and Berger (2002).

Additionally, using the above algorithm, we would like to know either the uncertainty of the variance estimate given the
245 number of Monte Carlo samples, or the number of samples required to obtain a particular level of Monte Carlo uncertainty on the variance. In essence, we would like to quantify the uncertainty of our uncertainty. To do so, we take a frequentist approach and construct confidence intervals on $\hat{\sigma}^2_\varphi$. The confidence intervals can be constructed by recognizing that the ratio of the Monte Carlo functional empirical posterior variance to the true functional posterior variance scaled by $(M-1)$ follows a $\chi^2_{M-1}$ distribution.

Since $\mathbf{c}^k_{MAP}$ is Gaussian, the random variables $\varphi_k = \boldsymbol{h}^\top \mathbf{c}^k_{MAP}$ $(k=1,\dots,M)$ are sampled independently and identically from a Gaussian distribution with some mean and variance $\sigma^2_\varphi = \boldsymbol{h}^\top \boldsymbol{\Sigma} \boldsymbol{h}$. By Theorem 5.3.1 of Casella and Berger (2002), we have the following distributional result,

$$\frac{(M-1)\hat{\sigma}^2_\varphi}{\sigma^2_\varphi} \sim \chi^2_{M-1}. \tag{20}$$

Thus, for $\alpha \in (0,1)$, the distribution in Equation (20) enables creating a $1-\alpha$ confidence interval for the true posterior variance,
$\sigma^2_\varphi$, as a function of the empirical posterior variance, $\hat{\sigma}^2_\varphi$. Using the exact distribution in Equation (20), we can create either one- or two-sided confidence intervals. Focusing on the two-sided case, we have,

$$\mathbb{P}\left\{ \chi^2_{M-1,\alpha/2} \leq \frac{(M-1)\hat{\sigma}^2_\varphi}{\sigma^2_\varphi} \leq \chi^2_{M-1,1-\alpha/2} \right\} = 1-\alpha, \tag{21}$$

where $\chi^2_{M-1,\alpha/2}$ is the $\alpha/2$-quantile of a chi-squared distribution with $M-1$ degrees of freedom. Hence, with some algebraic manipulation we arrive at the confidence interval of the posterior variance,

$$\mathbb{P}\left\{ \frac{(M-1)\hat{\sigma}^2_\varphi}{\chi^2_{M-1,1-\alpha/2}} \leq \sigma^2_\varphi \leq \frac{(M-1)\hat{\sigma}^2_\varphi}{\chi^2_{M-1,\alpha/2}} \right\} = 1-\alpha. \tag{22}$$

Since in practice we would like to characterize uncertainty in the same units as the flux estimate, we can provide an analogous confidence interval for the posterior standard deviation by taking square roots of all the terms within the probability statement in Equation (22), giving

$$\mathbb{P}\left\{\hat{\sigma}_\varphi\sqrt{\frac{M-1}{\chi^2_{M-1,1-\alpha/2}}}\leq\sigma_\varphi\leq\hat{\sigma}_\varphi\sqrt{\frac{M-1}{\chi^2_{M-1,\alpha/2}}}\right\}=1-\alpha. \tag{23}$$

Equation (23) facilitates the computation of a $(1-\alpha)\times100\%$ frequentist interval estimator of the Bayesian credible interval for the functional of interest $\varphi$. For each endpoint of the true Bayesian credible interval, we find a confidence interval such that the probability that both endpoint confidence intervals simultaneously cover the true credible interval endpoints is $1-\alpha$. Let $\gamma\in(0,1)$ and $\varphi_{MAP}=\boldsymbol{h}^\top\mathbf{c}_{MAP}$ be the functional MAP estimator as described in Section 2.1. Because the posterior is Gaussian and $\varphi_{MAP}$ is a linear functional, it is a one-dimensional Gaussian. Hence, the Bayesian $(1-\gamma)\times100\%$ credible interval is computed as follows,

$$\left[\underline{\varphi}^*,\overline{\varphi}^*\right]=\left[\varphi_{MAP}-z_{1-\gamma/2}\cdot\sigma_\varphi,\varphi_{MAP}+z_{1-\gamma/2}\cdot\sigma_\varphi\right], \tag{24}$$

where $z_{1-\gamma/2}$ is the $1-\gamma/2$ quantile of a standard Gaussian distribution. Equation (23) allows us to construct the aforementioned endpoint confidence intervals as follows. For readability, define $L^2:=\frac{M-1}{\chi^2_{M-1,1-\alpha/2}}$ and $R^2:=\frac{M-1}{\chi^2_{M-1,\alpha/2}}$. Thus, we have the following,

$$\begin{aligned}
1-\alpha&=\mathbb{P}\left\{z_{1-\gamma/2}\hat{\sigma}_\varphi L\leq z_{1-\gamma/2}\sigma_\varphi\leq z_{1-\gamma/2}\hat{\sigma}_\varphi R\right\}\\
&=\mathbb{P}\{-z_{1-\gamma/2}\hat{\sigma}_\varphi R\leq -z_{1-\gamma/2}\sigma_\varphi\leq -z_{1-\gamma/2}\hat{\sigma}_\varphi L\text{ and}\\
&\qquad z_{1-\gamma/2}\hat{\sigma}_\varphi L\leq z_{1-\gamma/2}\sigma_\varphi\leq z_{1-\gamma/2}\hat{\sigma}_\varphi R\}\\
&=\mathbb{P}\{\varphi_{MAP}-z_{1-\gamma/2}\hat{\sigma}_\varphi R\leq\varphi_{MAP}-z_{1-\gamma/2}\sigma_\varphi\leq\varphi_{MAP}-z_{1-\gamma/2}\hat{\sigma}_\varphi L\text{ and}\\
&\qquad\varphi_{MAP}+z_{1-\gamma/2}\hat{\sigma}_\varphi L\leq\varphi_{MAP}+z_{1-\gamma/2}\sigma_\varphi\leq\varphi_{MAP}+z_{1-\gamma/2}\hat{\sigma}_\varphi R\}\\
&=\mathbb{P}\{\varphi_{MAP}-z_{1-\gamma/2}\hat{\sigma}_\varphi R\leq\underline{\varphi}^*\leq\varphi_{MAP}-z_{1-\gamma/2}\hat{\sigma}_\varphi L\text{ and}\\
&\qquad\varphi_{MAP}+z_{1-\gamma/2}\hat{\sigma}_\varphi L\leq\overline{\varphi}^*\leq\varphi_{MAP}+z_{1-\gamma/2}\hat{\sigma}_\varphi R\}.
\end{aligned}$$

More concisely, defining

$$\underline{I}:=\left[\varphi_{MAP}-z_{1-\gamma/2}\hat{\sigma}_\varphi R,\varphi_{MAP}-z_{1-\gamma/2}\hat{\sigma}_\varphi L\right], \tag{25}$$

$$\overline{I}:=\left[\varphi_{MAP}+z_{1-\gamma/2}\hat{\sigma}_\varphi L,\varphi_{MAP}+z_{1-\gamma/2}\hat{\sigma}_\varphi R\right], \tag{26}$$

it follows that

$$\mathbb{P}\left\{\underline{\varphi}^*\in\underline{I}\text{ and }\overline{\varphi}^*\in\overline{I}\right\}=1-\alpha. \tag{27}$$

The intervals $\underline{I}$ and $\overline{I}$ quantify uncertainty on uncertainty, and provide a rigorous probabilistic characterization of the Monte Carlo procedure's uncertainty.

In practice, the original Bayesian credible interval in Equation (24) can be modified to account for the Monte Carlo uncertainty. To obtain an *upper bound* on the Bayesian credible interval, we apply an *inflation* factor (defined above as $R$), and thus obtain the interval $\left[\underline{\varphi}_u, \overline{\varphi}_u\right] = \left[\varphi_{MAP} - z_{1-\gamma/2}\hat{\sigma}_\varphi R, \varphi_{MAP} + z_{1-\gamma/2}\hat{\sigma}_\varphi R\right]$, such that $\mathbb{P}\left\{\left[\underline{\varphi}_u, \overline{\varphi}_u\right] \supset \left[\underline{\varphi}^*, \overline{\varphi}^*\right]\right\} = 1 - \alpha/2$. This probability is $(1 - \alpha/2)$ instead of $(1 - \alpha)$ since the probability in Equation (21) evaluated with only the lower bound yields a probability of $(1 - \alpha/2)$. Following the same steps as above, we obtain lower and upper bounds on the lower and upper endpoints of credible interval (24), respectively, holding with probability exactly $(1 - \alpha/2)$. Similarly, to obtain a *lower bound* on the Bayesian credible interval, we apply a *deflation* factor (defined above as $L$) and thus obtain the interval $\left[\underline{\varphi}_l, \overline{\varphi}_l\right] = \left[\varphi_{MAP} - z_{1-\gamma/2}\hat{\sigma}_\varphi L, \varphi_{MAP} + z_{1-\gamma/2}\hat{\sigma}_\varphi L\right]$ such that $\mathbb{P}\left\{\left[\underline{\varphi}_l, \overline{\varphi}_l\right] \subset \left[\underline{\varphi}^*, \overline{\varphi}^*\right]\right\} = 1 - \alpha/2$, holding with equality by the same logic as that used for the inflation factor.

Observing that the aforementioned inflation and deflation factors monotonically asymptote to one as the number of Monte Carlo samples $M$ gets large, the effect of the Monte Carlo procedure wanes as the number of samples grows. As shown in Table 2, the deflation factor monotonically approaches one from below as $M$ gets large while the inflation factor monotonically approaches one from above as $M$ gets large. As is characteristic of DA methods, each Monte Carlo iteration requires a non-trivial amount of computation, which practically restricts the number of Monte Carlo samples that can be obtained. As such, the inflated interval protects against underestimating the uncertainty, while the deflated interval provides a lower bound or "best-case" scenario for the uncertainty of the Bayesian procedure.

**Table 2.** Inflation and deflation factors for Monte Carlo (MC) estimated posterior standard deviation with $\alpha = 0.05$. When $M = 100$, by inflating the MC estimated posterior standard deviation by a factor of $1.1607$ (inflating by $16.07\%$), the extra uncertainty resulting from the MC procedure is accounted for with $97.5\%$ confidence. Similarly, when $M = 100$, deflating the MC estimated posterior standard deviation by a factor of $0.8785$ provides a lower bound on the true underlying Bayesian uncertainty with $97.5\%$ confidence. When considered simultaneously, the inflation and deflation factors bracket the true uncertainty with $95\%$ confidence.

| # Monte Carlo samples, $M$ | Deflation: $L = \sqrt{\frac{M-1}{\chi^2_{M-1,1-\alpha/2}}}$ | Inflation: $R = \sqrt{\frac{M-1}{\chi^2_{M-1,\alpha/2}}}$ |
| --- | --- | --- |
| 10 | 0.6987 | 1.7549 |
| 100 | 0.8785 | 1.1607 |
| 1,000 | 0.9580 | 1.0458 |
| 10,000 | 0.9863 | 1.0141 |
| 100,000 | 0.9956 | 1.0044 |
| 1,000,000 | 0.9986 | 1.0014 |

**Table 3.** Parameter settings for the low-dimensional example.

| Parameter | Value | Description |
|---|---|---|
| $\mathbf{c}_{true}$ | $\begin{bmatrix} 1 & 2 \end{bmatrix}^{\top}$ | True parameter vector |
| $\mathbf{c}_e$ | $\mathbf{c}_{true}$ | Expectation of the distribution in Equation (9) |
| $\mathbf{y}_e$ | $\mathbf{A}\mathbf{c}_{true}$ | Expectation of the distribution in Equation (10) |
| $b^2$ | 4 | The prior variance for each element in the parameter vector |
| $\sigma^2$ | 1 | Observation error variance |
| $m, n$ | 2 | Parameter and observation dimensions, respectively |
| $M$ | $10^2$ | The number of MC ensemble members |
| $\epsilon$ | 0.05 | Parameter of the matrix $\mathbf{A}$ defined in Equation (28) |

## 3 Numerical Examples

### 3.1 Low-Dimensional Example

We construct a two-dimensional example to provide a numerical demonstration that this MC procedure computes a consistent estimate of the posterior covariance, and is numerically close in practice. Define a linear forward model by,

$$\mathbf{A} = \begin{bmatrix} 1-\epsilon & \epsilon \\ \epsilon & 1-\epsilon \end{bmatrix}, \tag{28}$$

where $\epsilon > 0$. Let $\mathbf{c}_{true} \in \mathbb{R}^2$ be the true value of some physical parameter and suppose we set up Equations (5) and (6) as

$$\mathbf{c} \sim \mathcal{N}(\mathbf{c}_{true}, b^2 \mathbf{I}_2), \tag{29}$$

$$\mathbf{y} \mid \mathbf{c} \sim \mathcal{N}(\mathbf{A}\mathbf{c}, \sigma^2 \mathbf{I}_2). \tag{30}$$

We use the values in Table 3 to demonstrate the covariance agreement between the analytical equations for the posterior and the Monte Carlo ensemble member in addition to showing their agreement with the empirical covariance computed from the Monte Carlo ensemble members. We also provide an empirical demonstration that the empirical covariance matrix converges to the posterior covariance in the Frobenius norm as the number of ensemble elements $M$ gets large. Using the settings in Table 3, we obtain the following posterior covariance matrix by Equation (7):

$$\mathbf{\Sigma} = \begin{bmatrix} 0.87169811 & -0.07169811 \\ -0.07169811 & 0.87169811 \end{bmatrix}. \tag{31}$$

For the analytical covariance of the MAP estimator, we obtain the following matrix using Equation (12):

$$\mathbf{\Sigma}_{\mathbf{c}_{MAP}^k} = \begin{bmatrix} 0.87169811 & -0.07169811 \\ -0.07169811 & 0.87169811 \end{bmatrix}. \tag{32}$$

Indeed, these matrices are expected to be the same. Using simulated ensemble members, we obtain the following empirical covariance matrix using Equation (13):

$$\widehat{\boldsymbol{\Sigma}} = \begin{bmatrix} 0.8898093 & -0.22511958 \\ -0.22511958 & 1.03065003 \end{bmatrix}. \tag{33}$$

This empirical covariance matrix is qualitatively close to the analytical matrices, even for only $M = 10^2$. To show how the empirical covariance computed from the Monte Carlo ensemble elements converges to the true posterior covariance, we repeat the above implementation of the Monte Carlo procedure for $M = 100, 200, \ldots, 10^4$. For each $M$, we compute the Frobenius norm of the difference between the true posterior covariance and the empirical covariance to measure the estimation error, i.e., $\|\boldsymbol{\Sigma} - \widehat{\boldsymbol{\Sigma}}\|_F$. Since this estimation error is itself random, for each setting of $M$, we sample 100 realizations of the Monte Carlo procedure to obtain a mean norm difference. The results of this procedure are presented in Figure 1 and show that the estimation error exponentially decreases as the ensemble size gets large. By regressing $\log_{10}\|\boldsymbol{\Sigma} - \widehat{\boldsymbol{\Sigma}}\|_F$ on $\log_{10} M$, we estimate the error convergence rate to be $CM^{-0.49}$, where $C \approx 10^{0.22}$, as shown by the fitted line in Figure 1.

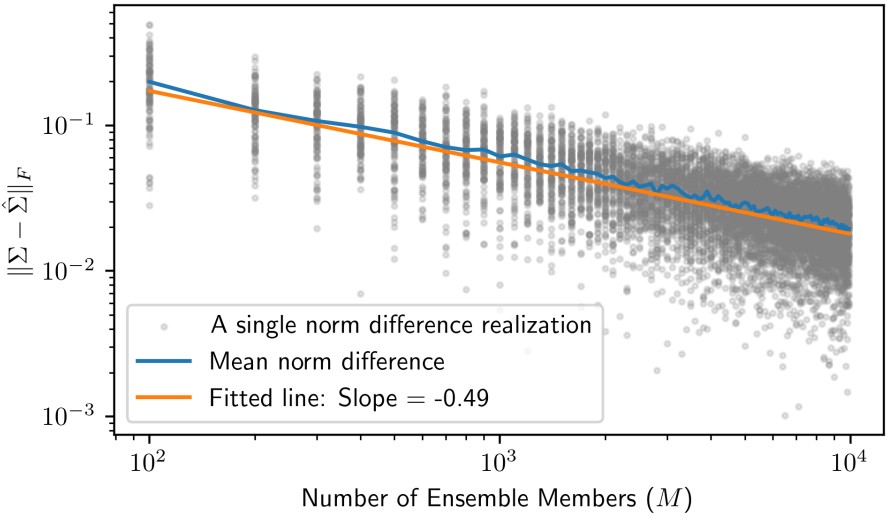

**Figure 1.** The Monte Carlo procedure is implemented on the low-dimensional example for a variety of ensemble sizes ($M = 100, 200, \ldots, 10^4$) to show that the empirical covariance resulting from the Monte Carlo procedure quickly gets close, in terms of Frobenius norm, to the posterior covariance as the number of ensemble members gets large. For each ensemble size $M$, we sample 100 realizations of the Monte Carlo procedure (shown in the gray dots) and find the average over these 100 realizations to see the expected behavior at each ensemble size. The orange regression line results from regressing $\log_{10}\|\boldsymbol{\Sigma} - \widehat{\boldsymbol{\Sigma}}\|_F$ on $\log_{10} M$, where the slope estimates the exponential error convergence rate.

## 3.2 Carbon Flux Inversion OSSE

We show an example of this Monte Carlo procedure being used to compute posterior uncertainties for global carbon fluxes along with the adjusted uncertainties from Section 2.3. We first detail adjustments to the 4D-Var setup and the Monte Carlo
procedure for carbon flux inversion, followed by the particulars of our simulation study and the Monte Carlo procedure results.

### 3.2.1 Applying 4D-Var and the Monte Carlo procedure to carbon flux inversion

Following along with the mathematical setup of Henze et al. (2007), the prior and posterior distributions are defined in a scaling factor space and hence the prior and posterior distributions on the physical quantity of interest are obtained by multiplying each respective scaling factor by a control quantity. In carbon flux estimation, the control quantity is a control flux, typically
an ansatz $CO_2$ flux between the Earth's surface and the atmosphere. Note that if the prior distribution mean in scaling factor space is unity, then the control flux is also the prior mean in the physical quantity of interest space. Mathematically, let $\mathbf{c} \in \mathbb{R}^m$ denote a scaling factor vector, $\tilde{\mathbf{y}} \in \mathbb{R}^n$ be the observation vector, $\boldsymbol{\mu} \in \mathbb{R}^m$ the control physical quantity, and $\boldsymbol{\theta} = \mathbf{c} \circ \boldsymbol{\mu} \in \mathbb{R}^m$ the physical quantity implied by $\mathbf{c}$ and $\boldsymbol{\mu}$, where $\circ$ denotes element-wise multiplication. We substitute these elements into the original mathematical model stated in Equation (1). The observation vector $\tilde{\mathbf{y}}$ is a sequence of $X_{CO_2}$ observations produced
by a remote sensing satellite, e.g., GOSAT or OCO-2 (O'Dell et al., 2012). The forward model, $f$, is a composition of an atmospheric transport model mapping scaling factors to atmospheric $CO_2$ concentrations with a remote sensing observation operator mapping $CO_2$ concentrations to $X_{CO_2}$ scalar values. The atmospheric transport model is known to be affine in the scaling factors due to the physics of $CO_2$ atmospheric transport. The exact mapping from atmospheric $CO_2$ concentrations to $X_{CO_2}$ is nonlinear, but in line with Liu et al. (2016), we use an affine approximation involving the known GOSAT averaging
kernel. As such, the affine composed function $f$ is of the form $f(\mathbf{c}; \boldsymbol{\mu}) = \mathbf{A}(\mathbf{c} \circ \boldsymbol{\mu}) + \mathbf{z}$, where $\mathbf{A} \in \mathbb{R}^{n \times m}$ is the linear forward model matrix, $\mathbf{c} \circ \boldsymbol{\mu}$ denotes the component-wise multiplication of $\boldsymbol{\mu}$ by the scaling factors $\mathbf{c}$ and $\mathbf{z}$ is comprised of the non-biospheric $CO_2$ contribution to the observations along with the prior mean of the $X_{CO_2}$ retrieval algorithm. As such, we define $\mathbf{y} := \tilde{\mathbf{y}} - \mathbf{z}$, giving the linear model

$$\mathbf{y} = \mathbf{A}(\mathbf{c} \circ \boldsymbol{\mu}) + \boldsymbol{\epsilon}, \quad \boldsymbol{\epsilon} \sim \mathcal{N}(\mathbf{0}, \mathbf{R}), \tag{34}$$

to which the previous analysis can nearly be applied. These definitions lead to the following Bayesian generative model,

$$\mathbf{c} \sim \mathcal{N}(\mathbf{c}^b, \mathbf{B}), \tag{35}$$
$$\mathbf{y} \,|\, \mathbf{c} \sim \mathcal{N}(\mathbf{A}(\mathbf{c} \circ \boldsymbol{\mu}), \mathbf{R}), \tag{36}$$

where $\mathbf{c}^b \in \mathbb{R}^m$ is the scaling factor prior mean and $\mathbf{B}$ is the prior covariance matrix.

In this study, the prior covariance $\mathbf{B}$ is parameterized with a single real value, $\mathbf{B} := b^2 \mathbf{I}_m$, where $b \in \mathbb{R}_+$. This is in line
with several published studies (Deng et al., 2014; Liu et al., 2016), and implies that all prior spatio-temporal indices are statistically independent. Similarly, the noise covariance $\mathbf{R}$ is assumed to be a diagonal matrix where each diagonal element is simply the variance of the corresponding $X_{CO_2}$ observation. As such, each diagonal element depends on the uncertainty of its corresponding $X_{CO_2}$ retrieval and the observations are assumed statistically independent given the scaling factors.

The uncertainty quantification objective is then to find the posterior uncertainty of a linear functional, defined by vector $\boldsymbol{h} \in \mathbb{R}^m$, of the physical quantity, $\boldsymbol{h}^\top \boldsymbol{\theta} = \boldsymbol{h}^\top (\mathbf{c} \circ \boldsymbol{\mu})$, given the observations. As shown in Algorithm (1), we can obtain an estimate of this uncertainty without computing the entire posterior covariance matrix as long as the covariance of the Monte Carlo ensemble elements is equal to the posterior covariance. We show this equivalence in the following argument.

For ease of notation, we rewrite Equation (36) using a short-hand notation for the $\circ$ operation (see Appendix A), i.e., $\mathbf{A}(\mathbf{c} \circ \boldsymbol{\mu}) = \mathbf{A}_{\boldsymbol{\mu}} \mathbf{c}$, so that

$$\mathbf{y} \mid \mathbf{c} \sim \mathcal{N}(\mathbf{A}_{\boldsymbol{\mu}} \mathbf{c}, \mathbf{R}). \tag{37}$$

Hence, applying Equations (7) and (8) and accounting for the control flux we have:

$$\boldsymbol{\Sigma} = \left( \frac{1}{b^2} \mathbf{I}_m + \mathbf{A}_{\boldsymbol{\mu}}^\top \mathbf{R}^{-1} \mathbf{A}_{\boldsymbol{\mu}} \right)^{-1} = \left( (\mathbf{A}^\top \mathbf{R}^{-1} \mathbf{A}) \circ \boldsymbol{\mu} \boldsymbol{\mu}^\top + \frac{1}{b^2} \mathbf{I}_m \right)^{-1}, \tag{38}$$

$$\boldsymbol{\alpha} = \boldsymbol{\Sigma} \left( \mathbf{A}_{\boldsymbol{\mu}}^\top \mathbf{R}^{-1} \mathbf{y} + \frac{1}{b^2} \mathbf{c}^b \right) = \boldsymbol{\Sigma} \left( (\mathbf{A}^\top \mathbf{R}^{-1} \mathbf{y}) \circ \boldsymbol{\mu} + \frac{1}{b^2} \mathbf{c}^b \right), \tag{39}$$

where Equation (38) follows from Corollary 4 and Equation (39) follows from Lemma 5 in Appendix A. These covariance and expectation equations define the Gaussian posterior for $\mathbf{c}$. The posterior distribution for the physical quantity, $\boldsymbol{\theta} = \mathbf{c} \circ \boldsymbol{\mu}$, is $\boldsymbol{\theta} \mid \mathbf{y} \sim \mathcal{N}(\boldsymbol{\delta}, \boldsymbol{\Gamma})$, where

$$\boldsymbol{\delta} = \mathbb{E}[\boldsymbol{\theta} \mid \mathbf{y}] = \mathbb{E}[\mathbf{c} \circ \boldsymbol{\mu} \mid \mathbf{y}] = \mathbb{E}[\mathbf{c} \mid \mathbf{y}] \circ \boldsymbol{\mu} = \boldsymbol{\alpha} \circ \boldsymbol{\mu}, \tag{40}$$

$$\boldsymbol{\Gamma} = \text{Cov}[\boldsymbol{\theta} \mid \mathbf{y}] = \text{Cov}[\mathbf{c} \circ \boldsymbol{\mu} \mid \mathbf{y}] = \text{Cov}[\mathbf{c} \mid \mathbf{y}] \circ \boldsymbol{\mu} \boldsymbol{\mu}^\top = \boldsymbol{\Sigma} \circ \boldsymbol{\mu} \boldsymbol{\mu}^\top, \tag{41}$$

where we have used Lemmas 1 and 2 from Appendix A.

The argument used in Equation (12) to establish the equivalence between the posterior covariance and the Monte Carlo ensemble element covariance also holds under the scaling factor and control flux setup in carbon flux inversion. One simply needs to apply the Lemmas in the Appendix to account for the component-wise relationship between the scaling factors and the control flux to establish covariance equality in the scaling factor space. This equality further extends to the physical quantity space, i.e., carbon flux space. The estimator of the physical quantity corresponding to $\mathbf{c}_{MAP}^k$ is

$$\boldsymbol{\theta}_k = \mathbf{c}_{MAP}^k \circ \boldsymbol{\mu}. \tag{42}$$

Using the result from Lemma 2 in Appendix A, the covariance matrix $\text{Cov}[\boldsymbol{\theta}_k]$ of this estimator is

$$\text{Cov}[\boldsymbol{\theta}_k] = \text{Cov}[\mathbf{c}_{MAP}^k \circ \boldsymbol{\mu}] = \text{Cov}[\mathbf{c}_{MAP}^k] \circ \boldsymbol{\mu} \boldsymbol{\mu}^\top = \boldsymbol{\Sigma} \circ \boldsymbol{\mu} \boldsymbol{\mu}^\top = \boldsymbol{\Gamma}. \tag{43}$$

Hence, the covariance matrix of the Monte Carlo physical quantity ensemble element is equal to the posterior covariance matrix of that physical quantity. Because of this equality, the empirical variance of the functional ensemble can be used to estimate the posterior variance of the functional of interest defined by $\boldsymbol{h}$, as shown in Step 3 of Algorithm (1). Note, because the validity of Algorithm (1) is based upon the covariance equivalence and not the functional of interest, once a Monte Carlo ensemble has been obtained, one can obtain posterior uncertainties for any collection of desired functionals without having

to generate a new Monte Carlo ensemble. This feature is particularly appealing in an application like carbon flux inversion, since scientists are usually interested in estimating posterior uncertainties for many regions (each of which can be encoded by a functional), which would be computationally cumbersome if each functional required a new Monte Carlo ensemble. It also enables obtaining uncertainties on new functionals that become subject of scientific interest only after the generation of the Monte Carlo ensemble.

### 3.2.2 OSSE Setup and Monte Carlo Procedure Results

We follow the flux inversion setup used by Byrne et al. (2019) (see Section 2.3 of that study). This setup uses the GEOS-Chem Adjoint model (Henze et al., 2007) to estimate scaling factors on a $4° \times 5°$ surface grid from January 2010 up to and including August. For each spatial point, there is one scaling factor parameter for each month, totaling $m = 72 \times 46 \times 8 = 26,492$ scaling factors parameters. This model is linear in terms of realistic fluxes (e.g., not including abnormally large negative fluxes), and hence amenable to this uncertainty quantification procedure. The OSSE defines ground-truth fluxes from the Joint UK Land Environment Simulator (JULES) (Clark et al., 2011; Harper et al., 2018) and uses Net Ecosystem Exchange (NEE) fluxes from NOAA's CarbonTracker version CT2016 (Peters et al. (2007), with updates documented at https://www.esrl.noaa.gov/gmd/ccgg/carbontracker/) as the control fluxes. The satellite $X_{CO_2}$ observations for the assimilation are generated from the JULES fluxes by running a forward GEOS-Chem simulation and sampling the model with the GOSAT observational coverage and observation operator (O'Dell et al., 2012).

The prior uncertainty, as described in Equation (5), is set to $b = 1.5$ (where $\mathbf{B} := b^2 \mathbf{I}_M$). To perform the Monte Carlo procedure, we draw $M = 60$ ensemble members, as described in Sec. 2.2. The $X_{CO_2}$ observation uncertainty $\boldsymbol{\Sigma}$ (a diagonal matrix, as the observations are assumed to be independent) comes directly from the GOSAT data product and varies between observations. For each ensemble member $k$, the output of the GEOS-Chem Adjoint optimization provides monthly scaling factor MAP estimators $\mathbf{c}^k_{MAP}$ according to the ensemble member inputs as described by Equation (11). Each ensemble MAP estimator is then multiplied by the control flux to obtain a MAP estimator in flux space.

The functionals of interest $\varphi$ are monthly global fluxes. The flux values on the 3-hour $4° \times 5°$ spatial-temporal grid are mapped to a global monthly flux using a weighted average with weights proportional to the surface area of each grid cell and uniform time weighting. The global flux posterior variance is computed for each month by finding the empirical variance of the Monte Carlo global flux members, as shown in Equation (18). To get a sense of how the DA is reducing prior uncertainty, for each month, we compute a % uncertainty reduction as follows:

$$\% \text{ Uncertainty Reduction} = 1 - \frac{\sigma_{posterior}}{\sigma_{prior}}. \tag{44}$$

Since we obtain an estimate of the posterior standard deviation through the Monte Carlo procedure, we do not precisely know Equation (44) and thus we consider the reduction both in terms of the raw Monte Carlo point estimate of the posterior standard deviation and its inflated version (i.e., $R$ as defined in Section 2.3).

The left side of Figure 2 shows the timeseries of global mean functionals and their credible intervals. The posterior flux is shown to have reduced error against the true flux, especially during the boreal summer months. Similarly, the Monte Carlo pos-

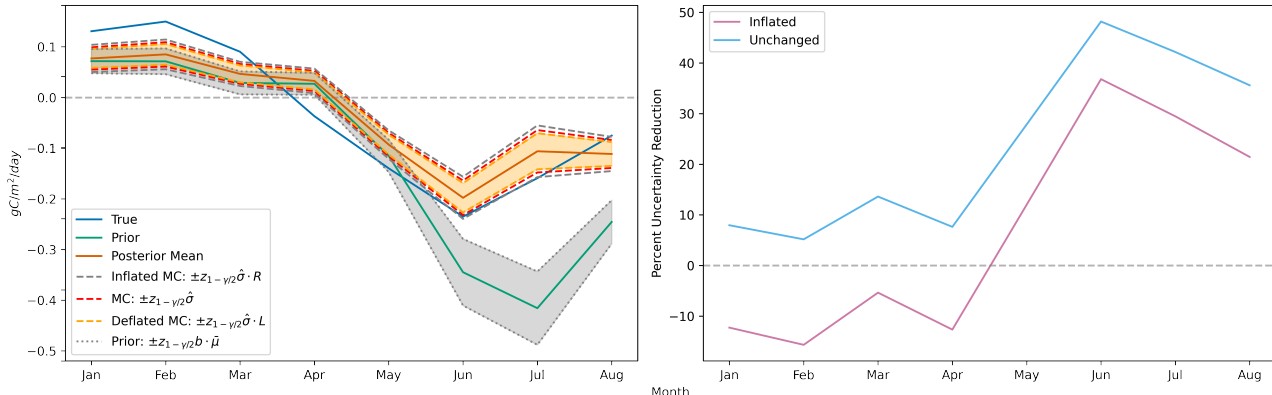

**Figure 2. (Left)** Estimated posterior $(1 - \gamma) \times 100\% = 95\%$ credible intervals around the monthly global flux functionals show markedly improved uncertainty over the prior during boreal summer months. The three interval types shown are the unchanged MC estimated intervals (red), the inflated MC estimated intervals (gray), and the deflated MC estimated intervals (orange). As described in Section 2.3, for each month, the true upper and lower credible interval endpoints are contained within the inflated and deflated endpoints with probability $1 - \alpha = 0.95$. Note, $\hat{\sigma}$ is a shorthand notation for the empirical functional standard deviation as defined in Step (3b) of Algorithm 1, $\overline{\mu}$ is the globally averaged control flux to which the prior uncertainty, $b$, is applied for each month, and $R$ and $L$ are the inflation and deflation factors, respectively, as defined in Table 2. We observe that with even as few as $M = 60$ ensemble members, at the monthly/global scale, the magnitude of the Monte Carlo sampling uncertainty is small in comparison to the posterior uncertainty. **(Right)** Percent reduction in uncertainty from prior to posterior for the monthly global fluxes is most significant during the boreal summer. The light blue curve shows the percent reduction estimated with the unchanged MC estimated posterior standard deviation, while the magenta curve shows the percent reduction estimated with the inflated MC estimated posterior standard deviation. The true reduction is larger than the reduction shown by the latter curve with $97.5\%$ confidence.

terior uncertainty estimate shows considerable reduction relative to the prior. The uncertainty estimates with inflated endpoints, increase the posterior uncertainty by $22\%$ while the deflated endpoints decrease the posterior uncertainty by $15\%$, resulting in credible interval endpoint bounds that capture the true credible interval endpoints with $95\%$ probability. The right side of Figure 2 further emphasizes the prior to posterior uncertainty reduction that we mathematically expect. However, we notice

that the inflated uncertainty is only reduced during boreal summer months. In January, February, March, and April the inflated Monte Carlo estimated posterior uncertainty is actually larger than the prior uncertainty. There is a logical explanation for this: since most of the landmass generating NEE fluxes is in the Northern Hemisphere and GOSAT requires sunlight to measure $X_{CO_2}$, the satellite observations impose much weaker constraints on the fluxes during boreal winter. Furthermore, since the prior uncertainty is defined as a percentage, the prior is more concentrated during the boreal winter months when the absolute

magnitude of the CarbonTracker fluxes is smaller. As a result of these two effects, the actual posterior uncertainty during the winter months is only slightly smaller than the prior uncertainty. Since we are obtaining a noisy Monte Carlo estimate of this

uncertainty from using 60 ensemble members, the inflated value accounting for the Monte Carlo uncertainty of the posterior uncertainty is slightly larger than the prior uncertainty.

## 4  Conclusions

For Bayesian uncertainty quantification in which the forward model is only available as a simulator, the carbon flux estimation community has proposed a useful Monte Carlo method to compute posterior uncertainties. This method is especially well-suited to DA tasks since it is parallelizable, works with computationally intensive physical simulators, and allows for flexible post-hoc uncertainty quantification on any desired functional of the model parameters. In this note, we analytically established the mathematical correctness of this procedure in the case of a linear forward model and Gaussian prior and error distributions

and provided additional uncertainty quantification to account for the Monte Carlo sampling variability in the final estimated credible interval. We also provided two numerical examples. In the first, we demonstrated the agreement between the analytical equations and empirical results for an explicitly known linear forward model. In the second, we showed that this procedure applies to a large-scale DA problem in the form of a carbon flux inversion OSSE, and reasoned that the uncertainty quantification results are mathematically and practically sensible.

Future investigations of this method could be based on an exploration of how many ensemble members must be sampled before the Monte Carlo uncertainty is sufficiently small in comparison to the posterior uncertainty. It is also not immediately clear if this procedure would work with DA algorithms other than 4D-Var and under a relaxation of the Gaussian assumptions as our demonstration relied upon explicitly showing the equivalence between the posterior and ensemble member covariances. As noted in Section 2.2, given this algorithm's similarity to the conditional sampling procedure in the spatial statistics literature,

there might be ways to relax the linear and Gaussian assumptions so that this algorithm can work in more general DA scenarios. Such an approach would need to demonstrate the covariance equivalence in some way. Finally, the validity of this procedure has been shown for quantifying the uncertainty in the model parameters, but there are also other sources of uncertainty in DA problems, such as uncertainty about the forward model or auxiliary inputs to the forward model. Since these uncertainty sources are typically quantified by looking at ensembles of models or model inputs, perhaps there is a way to use a Monte

Carlo procedure like this one to account for such systematic uncertainties as well.

## Appendix A:  Supporting Algebraic Results

There are a few key properties of the element-wise multiplication operation that must be stated in order to support the derivation of the equations presented in this paper.

For the following, let $\mathbf{x} \in \mathbb{R}^m$ be a random vector such that $\mathbb{E}[\mathbf{x}] = \boldsymbol{\mu}$ and $\mathrm{Cov}[\mathbf{x}] = \boldsymbol{\Sigma}$ and let $x_i$ denote the $i$th element of

465 $\mathbf{x}$. Additionally, suppose $\mathbf{a} \in \mathbb{R}^m$ and $\mathbf{A} \in \mathbb{R}^{n \times m}$.

**Lemma 1.** $\mathbb{E}[\mathbf{x} \circ \mathbf{a}] = \mathbb{E}[\mathbf{x}] \circ \mathbf{a}$.

*Proof.* By definition, we have

$$\mathbb{E}[\mathbf{x} \circ \mathbf{a}] = [\mathbb{E}[x_i a_i]]_i = [a_i \mathbb{E}[x_i]]_i = \mathbb{E}[\mathbf{x}] \circ \mathbf{a}. \tag{A1}$$

$\square$

**Lemma 2.** $\mathrm{Cov}[\mathbf{x} \circ \mathbf{a}] = \mathrm{Cov}[\mathbf{x}] \circ \mathbf{a}\mathbf{a}^\top$

*Proof.* There are two terms that need to be computed: (1) $\mathrm{Var}[x_i a_i]$ and (2) $\mathrm{Cov}[x_i a_i, x_j a_j]$. (1) is straightforward by properties of variance, namely, $\mathrm{Var}[x_i a_i] = a_i^2 \mathrm{Var}[x_i]$. (2) simply requires the definition of covariance, i.e.,

$$\mathrm{Cov}[x_i a_i, x_j a_j] = \mathbb{E}\big[(x_i a_i - \mathbb{E}[x_i a_i])(x_j a_j - \mathbb{E}[x_j a_j])\big] = a_i a_j \mathrm{Cov}[x_i, x_j]. \tag{A2}$$

Hence, it follows that $\mathrm{Cov}[\mathbf{x} \circ \mathbf{a}] = \mathrm{Cov}[\mathbf{x}] \circ \mathbf{a}\mathbf{a}^\top$. $\square$

**Lemma 3.** *Let $\boldsymbol{\mu} \in \mathbb{R}^m$ and let $\mathbf{A}_{\boldsymbol{\mu}}$ be such that the ith row of $\mathbf{A}_{\boldsymbol{\mu}}$ is equal to $[A_{ij}\mu_j]_i$. Then the following equation holds:*

$$\mathbf{A}_{\boldsymbol{\mu}}^\top \mathbf{A}_{\boldsymbol{\mu}} = \mathbf{A}^\top \mathbf{A} \circ \boldsymbol{\mu}\boldsymbol{\mu}^\top. \tag{A3}$$

*Proof.* To prove Equation (A3), first note that $\boldsymbol{\mu}\boldsymbol{\mu}^\top = [\mu_i \mu_j]_{ij}$. Let $i \in [m]$ and $j \in [n]$. By definition, we have

$$\big[\mathbf{A}_{\boldsymbol{\mu}}^\top \mathbf{A}_{\boldsymbol{\mu}}\big]_{ij} = \sum_{l=1}^m A_{li} A_{lj} \mu_i \mu_j = \mu_i \mu_j \sum_{l=1}^m A_{li} A_{lj}.$$

Hence, we can see that

$$\big[\mathbf{A}_{\boldsymbol{\mu}}^\top \mathbf{A}_{\boldsymbol{\mu}}\big]_{ij} = \left[\sum_{l=1}^m A_{li} A_{lj}\right]_{ij} \cdot [\mu_i \mu_j]_{ij}$$

and we have the desired result. $\square$

**Corollary 4.** *Let $\mathbf{M} \in \mathbb{R}^{n \times n}$ be a positive-definite matrix and let $\mathbf{A}_{\boldsymbol{\mu}}$ be defined as in Lemma 3. It follows that,*

$$\mathbf{A}_{\boldsymbol{\mu}}^\top \mathbf{M} \mathbf{A}_{\boldsymbol{\mu}} = \mathbf{A}^\top \mathbf{M} \mathbf{A} \circ \boldsymbol{\mu}\boldsymbol{\mu}^\top. \tag{A4}$$

*Proof.* Since $\mathbf{M}$ is positive-definite, it has a lower-triangular Cholesky decomposition, $\mathbf{M} = \mathbf{L}\mathbf{L}^\top$. For all $i \in [m]$ and $j \in [n]$, we have the following equivalence:

$$\big[\mathbf{L}^\top \mathbf{A}_{\boldsymbol{\mu}}\big]_{ij} = \sum_{l=1}^n A_{lj} L_{li} \mu_j = \Big[\big(\mathbf{L}^\top \mathbf{A}\big)_{\boldsymbol{\mu}}\Big]_{ij}. \tag{A5}$$

Therefore, $\mathbf{L}^\top \mathbf{A}_{\boldsymbol{\mu}} = \big(\mathbf{L}^\top \mathbf{A}\big)_{\boldsymbol{\mu}}$. Thus, Lemma 3 implies,

$$\mathbf{A}_{\boldsymbol{\mu}}^\top \mathbf{M} \mathbf{A}_{\boldsymbol{\mu}} = \big(\mathbf{L}^\top \mathbf{A}_{\boldsymbol{\mu}}\big)^\top \big(\mathbf{L}^\top \mathbf{A}_{\boldsymbol{\mu}}\big) = \big(\mathbf{L}^\top \mathbf{A}\big)_{\boldsymbol{\mu}}^\top \big(\mathbf{L}^\top \mathbf{A}\big)_{\boldsymbol{\mu}} = \big(\mathbf{L}^\top \mathbf{A}\big)^\top \big(\mathbf{L}^\top \mathbf{A}\big) \circ \boldsymbol{\mu}\boldsymbol{\mu}^\top, \tag{A6}$$

and the result follows since $\big(\mathbf{L}^\top \mathbf{A}\big)^\top \big(\mathbf{L}^\top \mathbf{A}\big) = \mathbf{A}^\top \mathbf{M} \mathbf{A}$. $\square$

**Lemma 5.** *Let $\mathbf{A}_{\boldsymbol{\mu}} \in \mathbb{R}^{n \times m}$ be defined as in the above, $\mathbf{M} \in \mathbb{R}^{n \times n}$ and $\mathbf{y} \in \mathbb{R}^n$. Then,*

$$\mathbf{A}_{\boldsymbol{\mu}}^{\top} \mathbf{M} \mathbf{y} = \mathbf{A}^{\top} \mathbf{M} \mathbf{y} \circ \boldsymbol{\mu}. \tag{A7}$$

*Proof.* It is sufficient to show the case when $\mathbf{M} = \mathbf{I}_n$, as otherwise we can simply define a new vector $\tilde{\mathbf{y}} = \mathbf{M}\mathbf{y}$. By matrix multiplication, for all $j \in [m]$,

$$\left[\mathbf{A}_{\boldsymbol{\mu}}^{\top} \mathbf{y}\right]_j = \sum_{i=1}^{n} A_{ij} y_i \mu_j = \left[\mathbf{A}^{\top} \mathbf{y} \circ \boldsymbol{\mu}\right]_j. \tag{A8}$$

$\square$

**Lemma 6.** *Let $\mathbf{A}_{\boldsymbol{\mu}} \in \mathbb{R}^{n \times m}$ be defined as in the above. Then,*

$$\mathbf{A}(\mathbf{a} \circ \boldsymbol{\mu}) = \mathbf{A}_{\boldsymbol{\mu}} \mathbf{a}. \tag{A9}$$

*Proof.* This property follows simply from the definition of $\mathbf{A}_{\boldsymbol{\mu}}$ and matrix multiplication. $\square$

**Corollary 7.** *Define $\mathbf{A} \in \mathbb{R}^{n \times m}$ as above, let $\alpha, \beta \in \mathbb{R}$, and let $\mathbf{a}, \mathbf{b}, \boldsymbol{\mu} \in \mathbb{R}^m$. Then $\mathbf{A}(\mathbf{a} \circ \boldsymbol{\mu})$ is linear in $\mathbf{a}$.*

*Proof.* By Lemma 6, we have,

$$\mathbf{A}\left((\alpha \mathbf{a} + \beta \mathbf{b}) \circ \boldsymbol{\mu}\right) = \mathbf{A}_{\boldsymbol{\mu}}\left(\alpha \mathbf{a} + \beta \mathbf{b}\right) = \alpha \mathbf{A}(\mathbf{a} \circ \boldsymbol{\mu}) + \beta \mathbf{A}(\mathbf{b} \circ \boldsymbol{\mu}). \tag{A10}$$

$\square$

*Author contributions.* MS and MK derived all mathematical and statistical results. BB and JL provided scientific expertise. BB provided the OSSE used for the numerical experiment. MS prepared the manuscript and ran all experiments with contributions from all co-authors. All authors participated in reviewing and editing the manuscript.

*Competing interests.* The authors declare that they have no competing interests.

*Acknowledgements.* This work was supported by NSF grant DMS-2053804, JPL RSAs No. 1670375, 1689177 & 1704914, and a grant from the C3.AI Digital Transformation Institute. Part of this research was carried out at the Jet Propulsion Laboratory, California Institute of Technology, under a contract with the National Aeronautics and Space Administration (grant no. 80NM0018D004). Liu and Byrne would like to acknowledge the funding support from NASA Orbiting Carbon Observatory Science Team program (17-OCO2-17-0013). Finally, we would like to thank Anna Harper for providing the JULES fluxes used in this study's OSSE, the STAMPS research group at Carnegie Mellon University for supporting this work, and the Uncertainty Quantification group at the Jet Propulsion Laboratory for facilitating this collaboration. We would like to acknowledge high-performance computing support from Cheyenne (doi:10.5065/D6RX99HX) provided by NCAR's Computational and Information Systems Laboratory, sponsored by the National Science Foundation.

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
