# Peer review of "Technical note: Posterior Uncertainty Estimation via a Monte Carlo Procedure Specialized for 4D-Var Data Assimilation"

_EGUsphere, 2023_

## Author Response (AR1)

Dear editor,

Thank you for considering our technical note, "Technical note: Posterior Uncertainty Estimation via a Monte Carlo Procedure Specialized for 4D-Var Data Assimilation." We are grateful for the helpful feedback from the two referees and below we have included our point-by-point responses to each of their comments. Each referee comment is included in bold and italicized text followed by our response and indications of changes made in our manuscript. Further, in our revised manuscript, all referee-inspired changes have been distinguished with blue text.

Sincerely,
Michael Stanley, Mikael Kuusela, Brendan Byrne and Junjie Liu

**Referee 1 Point-by-Point Responses**

***What the authors may not know is that the method, or something very similar to it, has already been used in the geostatistics field as a strategy for turning unconditional random field simulations into conditional simulations… This may also point towards generalisations of the method, because the original method was not constructed under a Gaussian assumption but it can still be used to estimate prediction covariances.***
Again, thank you for making this connection! This sampling method appears to be very similar to the one we discuss. In particular, both methods rely upon a covariance equivalence between the distribution of the conditional distribution of interest and the simulator to which one has access. As such, we have added your provided geostatistics references in Section 2.2, after articulating the necessity of the covariance equivalence. We also added a note in the Conclusion to articulate your point about this connection providing possible generalization beyond the Gaussian/linear assumptions. This is a good point, but we add that demonstrating the covariance equivalence would be the key to making this connection work in general. The Gaussian/linear assumptions allow us to prove this equivalence in this paper, but it is unclear how the proof would proceed otherwise.

***My main criticism is that the mathematical framework used is too strongly tied to flux inversion rather than a broader class of data assim[i]lation methods as promised in the title and abstract…Currently, the section is very tailored towards flux inversion, which I think does the generality of the method a disservice… An example of this is the use of the scaling factor parameterisation c o \mu, commonly used in flux inversion, which must be carried through the derivations but which has essentially nothing to do with the underlying Monte Carlo procedure. I think this section could be substantially simplified and made more general at the same time, and the details of the parameterisation could be deferred to Section 3.***
We agree that the original mathematical setup in Section 2 was too narrowly focused on the scaling factor specifics of the carbon flux inversion application. As such, we have reformulated Section 2 and the low-dimensional numerical example in Section 3 to not include the scaling factors. This reformulation has streamlined the mathematical presentation in Section 2 and defers the scaling factor formulation to Section 3, as suggested.
Regarding the applicability of this method to a broader class of data assimilation methods, we have adjusted the title and language throughout to specify that we are analyzing this method for 4D-Var data assimilation. Of course, understanding the broader applicability of this method to other data assimilation methods would be interesting, but we believe it is beyond the scope of this technical note.

***On the topic of generality, the simplification $B = b^2 I_m$ is introduced here that are not necessary for the mathematical results. Can this be left until section 3?***

Following the broader presentation adjustment of Section 2, we have moved this particular simplification to Section 3 for the carbon flux inversion example which uses this simplification.

*The situation in which the Monte Carlo method is applicable is the one where the posterior mean can be computed efficiently. In the flux inversion case, this arises due to the equivalence of the posterior mean and mode due to the Gaussian assumption, so that efficient gradient based methods that target the posterior mode can be used to compute the posterior mean (which one would typically compute analytically). The feasibility of this computation is critical to whether it's worth using this method, but as it stands this doesn't come out clearly in the note.*

To address the point of method assumptions and potential generalizations, we have added Section 2.2.1 (Considerations for when f is nonlinear and other critical assumptions). This section articulates the efficient computation assumption above in addition to the other Gaussian/linear assumptions and their necessity in our results. Regarding the linearity assumption, we provide two potential strategies to apply this method when the forward model is nonlinear.

*For the unconditional simulation the authors choose $y_k \sim N(A\mu, R)$ and, as the authors show, this gives the appropriate covariance to the samples $c_{MAP}^k$. However, Chevallier et al. (2007) took the mean of $y_k$ to be y, and as a consequence had $E(c_{MAP}^k) = \alpha$. In that case, the distribution of $c_{MAP}^k$ is precisely equal to the posterior distribution of c given y, which is elegant. In practice, any value can be chosen for the mean of y. There's no need to change this choice, but the difference to Chevallier and the freedom of this choice should be noted.*

Chevallier et al. (2007) indeed choose the mean of the unconditional observation distribution to be the "true" unperturbed observation (i.e., the forward model evaluated at the true state). We have added mention of this original choice in Section 2.2, but also make the point that since the the goal is to sample from a procedure with the correct covariance structure and this choice does not impact the covariance of the MAP estimator, this choice does not affect the validity of the uncertainties.

*Since it follows immediately from linearity, it may be worthing stating here that $c_{MAP}^k$ is Gaussian distributed, rather than deferring this to Section 2.3.*

We have moved this point that the MAP estimator is Gaussian to Section 2.2, directly after its definition (Equation 11).

**Referee 2 Point-by-Point Responses**

*My main concern is the strong focus on CO2 flux inversions. The method's derivation does not require the link to CO2 at all, thus I would appreciate a more general description of the method. E.g., in line 98 ff., the observations are described as $X_{CO2}$ retrievals although any observations should be feasible for the DA application. I suggest to re-write the section 2 by avoiding the link to CO2 flux inversions.*

We agree that the mathematical presentation in Section 2 was too strongly linked to the original carbon flux application. As such, we have reformulated Section 2 to a more general presentation by removing the setup based on scaling factors and references to the variables only in the carbon flux inversion context (e.g., the description of the observation vector). The rewrite has substantially streamlined the mathematical presentation and sharpened our main points.

*Further, a detailed discussion on the properties of the method is needed. For example, the 4D-var inversion method is described as it is important for the MC method. From*

*what I can see, any inversion method should be applicable in this context. Further, the requirements of the method remain unclear. Linear models and observation operators are applied in the demonstrator simulations as well as in the derivation of the method. In many field of application (e.g., NWP, air quality), this linear model is not applicable and the use of tangent-linear models is established. Would the method be also applicable in these areas?*

While it is possible that this Monte Carlo method is applicable to other data assimilation methods, we have changed the title and language throughout to specify that this technical note is discussing its applicability to 4D-Var data assimilation. However, this broader applicability is interesting and we have added a note about it in the conclusion. Within that context, we agree that the original version of this technical note lacked sufficient discussion around the method assumptions and properties. As such, we have added Section 2.2.1 (Considerations for when f is nonlinear and other critical assumptions) to address these points. In particular, we discuss some options of how this method can still be applicable when the forward model is nonlinear. We additionally included a discussion regarding the covariance equivalence property that must hold for this method to work and how there are potential generalizations that can be made via a similar sampling method from the geostatistics literature.

*Finally, I suggest to add a discussion on the ensemble type to which the method can be applied. E.g., does the method also work for multi-model or multi-parameter ensembles, which may contain more sources of uncertainty than only the flux scaling factors.*

The applicability of this method to other ensemble types to account for other types of uncertainty is a great question. Although we believe it is beyond the scope of this technical note to thoroughly  answer that question, we have added a brief note addressing it in the conclusion.

*[P]lease add a short description of particle filter methods, that provide uncertainty estimates without the need for linear model or Gaussian error statistic, in the introduction.*

Since we have amended this technical note to specifically address the scenario of UQ with 4D-Var data assimilation, we have addressed the possibility of relaxing the linear and Gaussian assumptions both by the inclusion of Section 2.2.1, and the connection to the geostatistics literature in Section 2.2. However, since we agree that mentioning particle filtering methods nicely rounds out the landscape of UQ methods for data assimilation, we have added a brief note in the Introduction on their utility in relaxing model and error assumptions.

*[L]ine: 24: put "infeasible" to the end of the sentence.*

This has been fixed.

*[L]ine 39: change "(Vasco et al., 1993)" to "Vasco et al. (1993)"; please revise the article and change other in-text citations accordingly.*

We have made this change and done a scan of the paper to look for other similar instances.

*[L]ine 68: The second statement on the UQ method (i.e., its independence on the prior biases) appears unproofed. Please include a dedicated paragraph to discuss the algorithms properties.*

We note in the derivation of Equation 12 that the choice of the prior mean does not affect the covariance of the MAP estimator, which is the primary equivalence required to make the method work correctly. This point is made explicitly in line 152 and is seen by the proof of the covariance equivalence.

***[L]ine 339: This is not clear to me. The derivation of the MC algorithm aims at defining a formula for the posterior uncertainty estimation. Why can't this estimate be used?***
The Monte Carlo procedure only provides an estimate of the posterior variance. The true posterior variance is unknown due to the forward model being only implicitly defined via a computational simulator (so it is for example not possible to evaluate Eq. (7) directly).

***[L]ine 367: The uncertainty obtained with the ensemble approach is an estimate of the posterior uncertainty. Thus, this statement is not clear to me. Why should the uncertainty of the approach be less than the posterior?***
Since there is uncertainty in our estimate of the posterior uncertainty from the Monte Carlo procedure, we ideally want to sample enough ensemble elements such that the uncertainty from the Monte Carlo procedure is sufficiently smaller than the actual posterior uncertainty. Otherwise, it is impossible to distinguish if the length of the resulting interval is primarily coming from the real posterior variance or the Monte Carlo uncertainty.

***line 375: please define [x]_i is element I of vector x/line 382: change [ ]_j to [ ]_i***
These changes have been made.

***Table 3: the number of ensemble members compared to the dimension of the problem is unrealistically large. Please provide a discussion on how the size of the ensemble improves the estimate of the MAP and its uncertainty. Specifically, how does the error in your estimate grow when taking only 10-100 samples of the prior PDF?***
We have addressed this in two ways. One, we have updated the experiment to have several orders of magnitude fewer samples to reduce the difference between the parameter dimension and the ensemble size. Two, we have included a new figure (Figure 1) showing how the discrepancy between the estimated covariance matrix and the true covariance matrix diminishes as a function of ensemble size.

---

## Author Response (AR2)

Dear Editor,

Thank you for passing along this final feedback. We have responded to the Referee #1's suggested edits below (the referee comment is bolded and italicized with our responses following in un-altered font) and have highlighted the changed text in blue in the corresponding updated manuscript.

***The title and the introduction restrict the scope of the note to 4D-Var, but this is not mentioned at all in the abstract. I would suggest slightly modifying the abstract to clarify this.***
We have added "4D-Var" to parts of the abstract describing the data assimilation task.

***Line 143: "Since the forward model A is only known implicitly via a computer simulator, making direct use of Equations (7) and (8) is intractable." It may be worth making this statement sharper by saying "numerically intractable", since its already been stated that it is analytically tractable.***
We appreciate the suggestion and have added "numerically" to be more precise.

***Lines 180-184: "As described in Chiles and Delfiner (2012), this sampling method is able to sample from complex conditional random fields if one has access to a simulator with the same covariance as the distribution of interest." More precisely, the method in question requires that unconditional simulations can be made from the appropriate distributions; the method then can turn these into conditional simulations.***
Thank you for the clarification. We have added some wording to more precisely make the connection.

***Line 195: the vector h here (and elsewhere) is bold but it is in italics, which is inconsistent with other vectors***
We have changed all instances of the h vector to be consistent with other vectors. All changes have been highlighted in blue.

***Table 1: the capitalisation is inconsistent in this table (e.g., "Control vector" in one line, "Observation Noise" in another)***
We have normalized the capitalization in Table 1 as suggested.

We again appreciate your feedback and consideration.

Sincerely,
Michael Stanley, Mikael Kuusela, Brendan Byrne and Junjie Liu